# BOOSTING IN-CONTEXT LEARNING IN LLMS WITH RETRIEVAL-BASED CODEBOOK

## ABSTRACT

Recent advancements in large language models (LLMs) have demonstrated exceptional performance across various downstream tasks, particularly due to their in-context learning (ICL) abilities. ICL enables models to learn from a few demonstrations presented in the context, without requiring retraining or fine-tuning. However, the effectiveness of ICL is highly dependent on factors such as prompt design and input length. To address these limitations, we propose a novel approach that leverages the key-value pairs within Transformers to enhance contextual understanding in LLMs. Specifically, our method converts raw demonstrations into task vectors—comprising keys and values—which are derived through multiple passes of the LLM, then integrated with test task vectors to improve model comprehension of the input. Furthermore, we introduce a retrieval-based codebook mechanism that captures information from long-context demonstrations while filtering irrelevant content. This codebook dynamically stores and updates task vectors generated during inference, mitigating input length constraints and optimizing the relevance of contextual data. By retrieving the most pertinent historical task vectors, the codebook ensures that only relevant information is utilized during inference. Extensive experiments show that these enhancements significantly outperform conventional ICL, achieving superior accuracy and efficiency. Overall, this work sets a new benchmark for optimizing ICL in LLMs, enabling their effective deployment in complex, real-world applications.

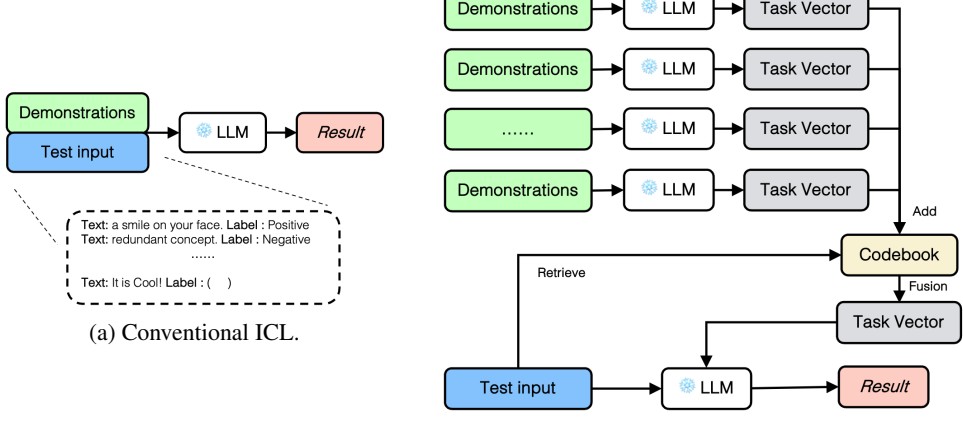

(a) Conventional ICL.

(b) Boosting ICL with retrieval-based codebook.

Figure 1: Intuitively compare conventional ICL with ours.

## 1 INTRODUCTION

Recently, large language models (LLMs) have shown excellent performance across a wide range of downstream tasks Zhao et al. (2023), such as commonsense question answering Bian et al. (2024), fact verification Tang et al. (2024), and natural language inference Qiao et al. (2023). During their

application in various domains, many studies have found that LLMs exhibit strong in-context learning (ICL) capabilities Dong et al. (2022). This means they can learn from a few demonstrations within the input context and effectively perform different tasks without requiring retraining or fine-tuning of model parameters. However, the performance of ICL is influenced by complex factors Dong et al. (2022). While downstream task accuracy is a key metric, conventional ICL often underperforms due to suboptimal prompt settings Liu et al. (2024a). Additionally, the ability of LLMs to handle long-context inputs plays an important role, as input length constraints can limit their ability to effectively learn from demonstrations Li et al. (2024).

Since ICL performance is highly sensitive to prompt settings and other factors, enhancing its efficacy is crucial. Prompts typically consist of a query and demonstration context written in natural language and are fed into LLMs for prediction Wang et al. (2020). These characteristics make ICL well-suited for human interaction. Previous work on enhancing ICL has primarily focused on improving prompt design, including the selection and ordering of demonstrations as well as instruction formatting Wang et al. (2023). Selecting suitable examples aims to improve ICL performance, while the order in which demonstrations are presented also significantly impacts model comprehension.

As ICL is a relatively new paradigm, its underlying mechanisms remain uncertain, making prompt engineering unstable Dai et al. (2023). To enhance ICL performance effectively without additional training, we propose a novel ICL enhancement method. We posit that demonstrations input into LLMs are transformed into high-dimensional vectors or representations. The key-value pairs of Transformers across each layer serve as suitable process variables, as Transformers are the foundational components of LLMs, encoding the task paradigms necessary for understanding the input during inference. Simultaneously, considering classic residual methods, we hypothesize that raw demonstrations still contain valuable contextual information. Therefore, these demonstrations are reintroduced as input after initial comprehension. Specifically, when the key-value pairs are extracted, they are concatenated with those derived during the repeated processing of the raw demonstrations. This iterative process allows the model to better comprehend the context than through a single pass.

Another challenge in ICL is managing input length constraints and noise. In certain LLMs, especially those without position embedding strategies like RoPE Su et al. (2024) or other length-expanding methods Xiong et al. (2024), long-context or large demonstrations cannot be effectively processed, impairing comprehension. Additionally, when demonstrations are lengthy, irrelevant content and noise within the context can degrade ICL performance. To address this, we propose a retrieval-inspired mechanism Lewis et al. (2020) for key-value pairs. We introduce a codebook Hartvigsen et al. (2023)—a modifiable memory structure that stores key-value pairs from demonstrations processed multiple times by the LLM. This codebook retains all demonstration information while allowing obsolete content to be updated, edited, and revised, ensuring only relevant memory is utilized by the LLM. When a test query is input, the most useful, similar, and relevant key-value pairs are retrieved from the codebook. These retrieved pairs capture the aspects most likely to enhance ICL performance and play a crucial role in overcoming long-context limitations. The retrieved and refined representations serve as enhanced prompts for the test input.

In summary, this paper makes the following contributions: (1) We investigate and address limitations in ICL by introducing techniques that optimize prompt design and improve the utilization of Transformer key-value pairs, enhancing contextual understanding in LLMs for a range of downstream tasks. (2) We propose a retrieval-inspired mechanism that uses a dynamic codebook to manage key-value pairs generated over multiple passes, effectively overcoming input length constraints and filtering irrelevant information to improve inference relevance. (3) Through extensive experiments, we demonstrate that our enhancements outperform state-of-the-art ICL methodologies in both accuracy and efficiency. This work sets a new benchmark for optimizing ICL in large language models, paving the way for their effective deployment in complex, real-world applications.

## 2 RELATED WORK

### 2.1 KEYS AND VALUES IN LLMS

Keys, values, and queries are crucial components in the self-attention mechanisms that form the backbone of Transformers and LLMs. During the inference phase, keys and values serve as rel-

atively fixed variables, encapsulating high-dimensional features of demonstrations and remaining unaffected by input length constraints. Previous studies have suggested that ICL can be viewed as compressing a training set into a single task vector Hendel et al. (2023), essentially another form of high-dimensional feature representation. This viewpoint highlights the importance of extracting keys and values effectively. Moreover, in an effort to emulate human cognitive processes, methodologies like Deep-thinking Yang et al. (2024b) enhance keys and values by iteratively processing demonstrations, refining their understanding through multiple passes. The KV cache is another widely adopted technique that leverages the length-insensitive nature of compressed data to accelerate inference Liu et al. (2024b). However, while these approaches focus on enhancing computational efficiency, there remains a notable gap in integrating the interpretability and utility of keys and values directly within the ICL framework Hooper et al. (2024).

## 2.2 DEMONSTRATION DESIGN

In ICL, demonstration inputs are combined with test inputs into a single context for the LLM. The model then uses these demonstrations to make predictions for the test inputs, effectively transferring classification and answering skills from the given examples. Despite the potential of ICL, research into its variants and enhancement methods has been limited. Demonstration design plays a pivotal role during the ICL inference stage, as it can significantly influence model performance Lu et al. (2022a). Past work has concentrated on selecting and ordering raw demonstrations to optimize their utility, determining both which examples best support ICL and in what demonstrations they should be presented Dong et al. (2022). Common selection techniques often rely on established distance metrics, information theory, and computational linguistics to identify "closest neighbors" Qin et al. (2023); Liu et al. (2022); Sorensen et al. (2022); Gonen et al. (2023). However, this approach can sometimes overlook the nuanced understanding that LLMs inherently possess and may treat the selection process as an isolated embedding module separate from the ICL framework. Considering the robustness of LLMs as inference tools, this reliance on external selection mechanisms can be questioned. Moreover, research shows that the organization of demonstrations impacts ICL performance, leading to efforts to reorder demonstrations based on their relationship to the input Lu et al. (2022b). However, this reordering is often complex and may not yield optimal results. As such, we posit that the presentation order may be less critical when demonstrations are fully encapsulated within the keys and values across LLM layers, allowing the model to utilize multi-layered contextual understanding without depending heavily on sequence.

## 2.3 CODEBOOK

A codebook is an abstract storage concept, typically associated with vectors but encompassing a variety of storage, compression, and editing techniques. Codebooks have been employed for knowledge editing Hartvigsen et al. (2023), functioning as repositories for both outdated and newly acquired knowledge. Furthermore, in specific scenarios, codebooks provide standardized storage formats for label assumptions that LLMs must respect during text generation. Recent designs, such as the LLM-codebook Deng et al. (2024), map extended language models into compressed codebooks to enhance model efficiency and reduce size. Additionally, in multimodal tasks, codebooks serve as generalization standards, as seen in the context of Unicode Zheng et al. (2024). While the concept of codebooks is highly abstract and versatile, within the scope of our research, their role is more aligned with knowledge editing. Specifically, the codebook acts as a repository for effectively understanding and storing historical demonstrations, serving as a refined memory structure to improve the relevance and utility of contextual information during ICL inference.

## 3 OUR PROPOSAL

### 3.1 BACKGROUND

In-context learning is the problem to solve in our work. The input of ICL consist of two part: demonstrations input $X_{demos}$ and test input $X_{test}$, where $X_{demos} = \{x_i, y_i\}_{i=1}^{S}$ and $X_{test} = \{x_{test}\}$. $S$ means S-shot in ICL, if there is a 10 classification task, $S$ is a multiple of 10. ICL aims to predict $X_{test}$ label $\hat{y}$ from $Y$, which is the set of list $\{y_1, y_2, ..., y_S\}$. From view of calculating

process of LLM $M$,

$$\hat{y} = \arg\max_{y_j \in Y} P_M(y_j | X_{demos}, x_{test}), \tag{1}$$

where $P$ is the output logits of $M$.

## 3.2 OVERVIEW

As shown in Figure 2, the overall framework of the proposed method mainly consists of two parts. The first part involves multiple reflections on demonstrations and the calculation of the final results. The second part is about the operations related to the codebook, mainly the three basic running functions of the codebook.

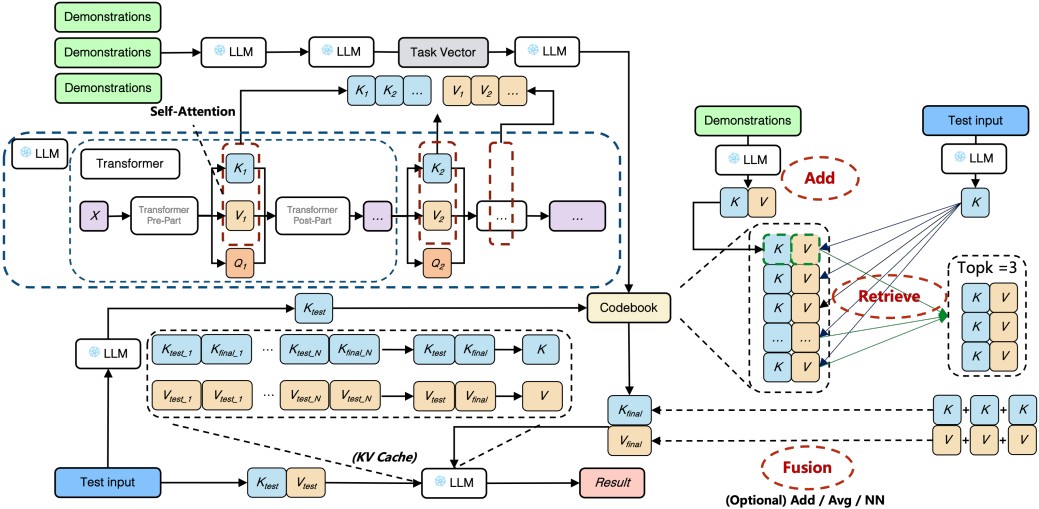

Figure 2: Overview of boosting in-context learning through retrieval-based codebook.

## 3.3 METHODOLOGY

**Learning Algorithm $A$ and Rule Application $f$.** To understand the mechanism behind ICL, previous research has proposed a universal theoretical framework based on learning theory from the perspective of hypothesis classes Hendel et al. (2023). In this framework, the fundamental components remain consistent: the decoder-only LLM $M$, which consists of a Transformer with $N$ layers, and the inputs and outputs of ICL, denoted as $X_{\text{demos}}$ and $X_{\text{test}}$. This theoretical framework can be divided into two main components: the learning algorithm $A$, which maps $X_{\text{demos}}$ into a task vector, and the rule application $f$, which maps the query $X_{\text{test}}$ into an output based on the task vector. Within this framework, ICL can be summarized by the following formula:

$$M\left([X_{\text{demos}}, X_{\text{test}}]\right) = f\left(x; A\left(X_{\text{demos}}\right)\right). \tag{2}$$

The generality of this theoretical framework is evident in its various implementations, which depend on the specific forms or structures of the chosen learning algorithm $A$ and rule application $f$.

For general customization, building on previous work, we propose using the keys and values of the Transformer as the output of the mapping of $X_{\text{demos}}$ through the learning algorithm $A$. The attention weights of the $n$-th Transformer layer are computed as follows:

$$K_n = W_K X_{n-1}, \quad Q_n = W_Q X_{n-1}, \quad V_n = W_V X_{n-1}. \tag{3}$$

The LLM $M$, which consists of $L$ layers of Transformers, produces $L$ pairs of keys and values from the attention mechanism of each layer. The keys and values represent the high-dimensional features of $X_{\text{demos}}$. The learning algorithm $A$ can be viewed as the process that computes the keys and values within the Transformer architecture based on $X_{\text{demos}}$:

$$A : A_{\text{single}} = \{\{K_i\}_{i=1}^L, \{V_i\}_{i=1}^L\} = \{\{K_1, K_2, \ldots, K_L\}, \{V_1, V_2, \ldots, V_L\}\} = \{K_A, V_A\} \tag{4}$$

In summary, $A_{\text{single}}$ generates a task vector for the testing process. The testing process of ICL is calculated as follows:

$$K_{\text{test}} = W_K X_{\text{test}}, Q_{\text{test}} = W_Q X_{\text{test}}, V_{\text{test}} = W_V X_{\text{test}},$$
$$f : \text{Output} = \text{Attention}(\{K_{\text{test}}\|K_L\}, \{V_{\text{test}}\|V_L\}, Q_{\text{test}}). \tag{5}$$

This design allows for a more flexible combination of the learning algorithm $A$ and the rule application $f$, providing opportunities for improvement in both areas. The above is shown in Figure 3.

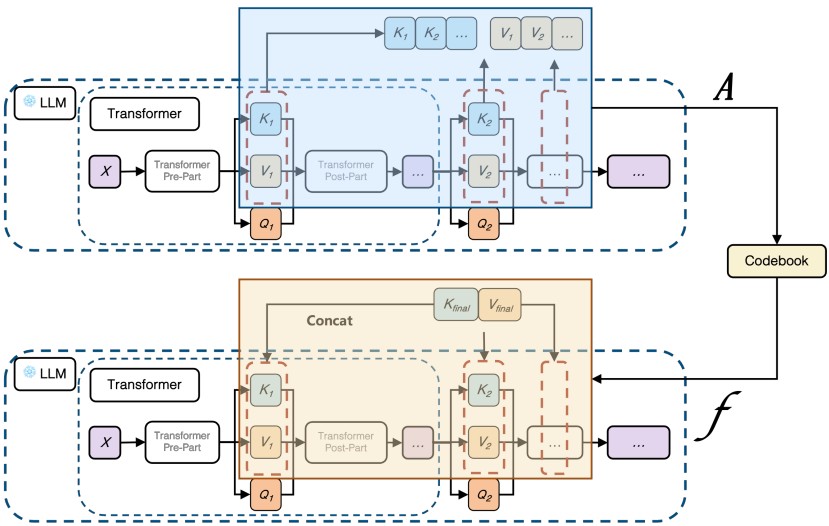

Figure 3: Extracting keys and values as the perspective of hypothesis classes.

**Multiple Boosting of the Task Vector.** The task vector $A_{\text{single}}$, obtained from a single learning algorithm $A$, raises the question of whether it can be further enhanced to achieve better results during the testing process, particularly when applying $f$. The task vector generated by the learning algorithm $A$ exists in the form of keys and values, indicating it can be reused during the computations of the LLMs. Thus, it can indeed be recomputed (or reintegrated) by the LLMs. For each LLM, the calculation process that concatenates the previous task vector follows:

$$K_{\text{demos}} = W_K X_{\text{demos}}, Q_{\text{demos}} = W_Q X_{\text{demos}}, V_{\text{demos}} = W_V X_{\text{demos}},$$
$$\text{Output}_M = \text{Attention}(\{K_{\text{demos}}\|K_{\text{past}}\}, \{V_{\text{demos}}\|V_{\text{past}}\}, Q_{\text{demos}}). \tag{6}$$

Here, the variable containing demonstrations signifies that the LLM re-evaluates the raw demonstrations (similar to a residual connection) while accepting the past task vector. $\text{Output}_M$ represents the output of the LLM based on the past keys and values $K_{\text{past}}$ and $V_{\text{past}}$, which are the task vectors from prior LLM evaluations.

The previous and newly task vectors, arising from the re-evaluation of the demonstrations, serve as two computational components in the overall process. They aim to achieve two objectives: enhancing the re-evaluation of the demonstrations, which relates to the depth dimension of the LLM layers—reflecting the single computation process—and leveraging past task vectors to improve the new task vector's quality. To this end, we stack several LLMs to iteratively enhance the task vector, thereby creating a new task vector to pass to the subsequent LLM. By introducing a decay rate $\eta$, we can maintain a balance between the past and present task vectors:

$$K_{\text{present}} = \eta K_{\text{demos}} + (1 - \eta) K_{\text{past}}$$
$$V_{\text{present}} = \eta V_{\text{demos}} + (1 - \eta) V_{\text{past}} \tag{7}$$
$$A_{\text{present}} = \{K_{\text{present}}, V_{\text{present}}\}$$

Through this $N$ $L$-layer LLM enhancement method, we finally derive the task vector for $f$.

**Retrieve-Based Codebook.** To address the limitations posed by the number of demonstrations, especially when the number of demonstrations $S$ in the input $X_{\text{demos}}$ becomes too large for the

LLMs to handle due to the constraints of positional embedding methods (which are not RoPE or other length-expanding methods), we replace $X_{\text{demos}}$ with:

$$X_{\text{codebook}} = \{X_{\text{demos}_1}, X_{\text{demos}_2}, \dots, X_{\text{demos}_C}\},\tag{8}$$

where $c$ denotes the number of items in the codebook, achieved through either splitting or adding new demonstrations. Each element in $X_{\text{codebook}}$ undergoes multiple boosting processes:

$$\{A_i\}_{i=1}^C = \{K_{A_i}, V_{A_i}\}_{i=1}^C\tag{9}$$

where $A_i$ is computed as in equations (3) and (4). Each $A_i$ represents the boosted understanding of the task vector and consists of keys and values from $N$ layers of the LLM $M$.

Before inputting the first demonstration into the LLM, a discrete codebook $CB$ exists outside the LLM's computation process. This codebook contains two components: Keys (K) and Values (V), which are structured as follows. The task vectors (keys and values from $N$ layers) of each demonstration are stored in $CB$:

$$CB = \{A_1, A_2, \dots, A_C\},\tag{10}$$

where $A_i$ is defined according to (9). From the perspective of knowledge editing, $CB$ is both editable and updatable. If historical demonstrations are outdated or incorrect, they must be removed or corrected; if new demonstrations arise, they should be added to $CB$. We have implemented dynamic additions to $CB$. However, since knowledge editing is not the focus of this article, the functional components for editing outdated information have not been implemented, nor have their effects been evaluated.

After the test input $X_{\text{demos}}$ is processed multiple times, yielding the task vector $A_{\text{demos}}$, we calculate the similarity between $K_{\text{demos}}$ and every key $A_i$ in $CB$. The method for similarity calculation is flexible; options include cosine similarity, Euclidean distance, and more. We introduce a hyperparameter $T$ to denote the number of results to return after retrieval. We select the $T$ task vectors $A_i$ that exhibit the highest similarity as the retrieval results $C_r$:

$$C_r = \{A_i\}_{i=1}^T.\tag{11}$$

Next, we employ a fusion method fusion to merge the retrieval results, which can adopt various approaches including summation, averaging, or using a trainable network, ultimately yielding a unified output $A_{\text{final}}$:

$$A_{\text{final}} = \text{fusion}(C_r).\tag{12}$$

Finally, the resulting task vector $A_{\text{final}}$ is concatenated to produce the final output:

$$\begin{aligned} A_{\text{final}} &= \{K_{\text{final}}, V_{\text{final}}\} \\ \text{Output}_M &= \text{Attention}(\{K_{\text{test}}\|K_{\text{final}}\}, \{V_{\text{test}}\|V_{\text{final}}\}, Q_{\text{test}}). \end{aligned}\tag{13}$$

Drawing from numerous historical demonstrations, we seamlessly integrate the functions of addition, retrieval, and fusion to ultimately achieve the output of ICL, $\text{Output}_M$.

## 4 EVALUATION

### 4.1 SETUP

**Datasets and Baselines.** To assess the effectiveness of our proposed method, we evaluate its performance alongside conventional ICL on several widely used datasets: SST2 Socher et al. (2013), SST5 Socher et al. (2013), MR Pang & Lee (2005), and AGNews Zhang et al. (2015). The evaluations are performed using LLMs of various sizes, including opt-125m, opt-350m Zhang et al. (2022), Qwen2-1.5B, Qwen2-7B Yang et al. (2024a), and Llama3.1-8B Dubey et al. (2024). Table 1 provides a summary of the key characteristics of these datasets, including the size of the validation set, maximum text length, and domain. We also used a private dataset within Ant Group called AE for testing. This is a 28 category merchant name industry classification dataset.

**Implementation Details.** All experiments were conducted using Python 3.8, PyTorch 2.1 Paszke et al. (2019), and transformers 4.43 Wolf et al. (2020), along with compatible auxiliary libraries. The computational resources used include a single NVIDIA Tesla A100 GPU with 80 GB memory. In our setup, the number of task vectors in the codebook $C$ is set to 10 (Equation 10), and the

Table 1: Dataset statistics.

| Dataset | Categories | Size of validation | Max text length | Domain |
|---------|-----------|--------------------|-----------------|--------|
| SST2 | 2 | 872 | 65 | Sentiment |
| SST5 | 2 | 1101 | 65 | Sentiment |
| MR | 2 | 1066 | 68 | Comment |
| AGNews | 4 | 7600 | 217 | News |
| AE | 28 | 1000 | 116 | Industry |

number of task vectors selected for fusion $T$ is set to 5 (Equation 11). To balance time consumption and performance, we enhance the task vectors by LLMs, where the number of LLMs $N = 5$. Model performance is evaluated based on the accuracy of ICL in completing classification tasks. For clarity and reproducibility, we provide pseudocode outlining the computational reasoning, as shown in Algorithm 1.

---

**Algorithm 1** Overall pseudocode.

---

**Require:** Demonstrations $X_{demos}$; test input $X_{test}$; $N$ transformer-based LLM $M$ with $L$ layers; codebook $CB$; the number of items in the codebook $C$; topK selection $T$;
1: **for** $X_{demos\_c} \in X_{codebook}$ **do**
2:     Initialize $X_0 = X_{demos\_c}$
3:     **for** $n \in N$ **do**
4:         Initialize $X_0 = X_{l-1}$ if $l - 1 >= 0$ else $X_0$
5:         **for** $l \in L$ **do**
6:             $Q_l, K_l, V_l = (W_{lq}, W_{lk}, W_{lv})X_l$
7:             $X_{l+1} = Attention(Q_l, K_l, V_l)$
8:         **end for**
9:         $X_n = X_L$
10:     **end for**
11:     $A_n = \{\{K_i\}_{i=1}^L, \{V_i\}_{i=1}^L\}$
12:     $CB.insert(A_n)$
13: **end for**
14: **for** $c \in C$ **do**
15:     $C_r = topk(CB_c.\{K_i\}_{i=1}^L, X_{test}.\{K_i\}_{i=1}^L)$
16: **end for**
17: Initialize $X_0 = X_{test}$
18: **for** $n \in N$ **do**
19:     $X_n = X_{n-1}$
20:     **for** $l \in L$ **do**
21:         $Q_l, K_l, V_l = (W_{lq}, W_{lk}, W_{lv})X_l$
22:         $X_{l+1} = Attention(Q_l, K_l, V_l)$
23:     **end for**
24:     $X_n = X_L$
25: **end for**
26: $A_{test} = \{\{K_i\}_{i=1}^L, \{V_i\}_{i=1}^L\}$
27: $C_r = topk([CB, A_{test}])$
28: $A_{final} = fusion(C_r) = \{K_{final}, V_{final}\}$
29: $Output_M = \text{Attention}(\{K_{test}\|K_{final}\}, \{V_{test}\|V_{final}\}, Q_{test})$
30: **return** $Output_M$

---

### 4.2 MAIN RESULT

Table 2 presents the main results of our method across the four selected datasets. Compared to conventional ICL, which processes the input only once, our approach achieves significantly improved accuracy. Moreover, we observe that model performance generally improves as the parameter size of the LLMs increases, indicating a positive correlation. The performance gain is particularly pro-

nounced for LLMs with smaller parameter sizes, as larger models already demonstrate strong results in ICL tasks, leaving less room for improvement. It is noteworthy, however, that in certain cases, the relationship between parameter size and performance does not follow a strictly positive correlation. This discrepancy is primarily due to variations in quantization strategies. Specifically, while we utilized 8-bit quantization for the Opt and Llama3.1, the Qwen2 was left unquantized due to its adaptive parameter quantization approach. Despite these differences, the results consistently demonstrate the effectiveness of our method across various LLMs. Evaluating different LLMs not only highlights the benefits of increasing parameter counts but also confirms the robustness of our method when applied to LLMs trained on different foundations and pretraining techniques.

Table 2: Main results of conventional ICL and ours across different model on selected datasets.

| Model | Method | SST2 | SST5 | MR | AGNews |
|---|---|---|---|---|---|
| OPT-125M | ICL | 55.43 | 18.46 | 48.19 | 49.37 |
|  | *Ours* | **77.98** | **22.62** | **60.79** | **63.75** |
| OPT-350M | ICL | 58.36 | 20.98 | 49.47 | 54.91 |
|  | *Ours* | **81.08** | **25.15** | **63.32** | **69.25** |
| Qwen2-1.5B | ICL | 57.13 | 19.03 | 48.46 | 52.04 |
|  | *Ours* | **62.39** | **27.98** | **60.32** | **61.65** |
| Qwen2-7B | ICL | 81.95 | 25.64 | 58.05 | 59.43 |
|  | *Ours* | **87.61** | **31.97** | **65.29** | **83.30** |
| Llama3.1-8B | ICL | 82.10 | 27.39 | 60.53 | 60.19 |
|  | *Ours* | **91.32** | **29.41** | **68.39** | **88.96** |

For the AE dataset, which contains 28 categories, we directly employed larger LLMs, including Qwen2-7B, Qwen2-7B-Instruct, Llama3.1-8B, and Llama3.1-8B-Instruct, for evaluation. Additionally, we compared our method against other ICL enhancement baselines. The results indicate that our model outperforms both the other ICL baselines and the conventional ICL Yang et al. (2024b) on the AE dataset. The results are presented in Figure 4.

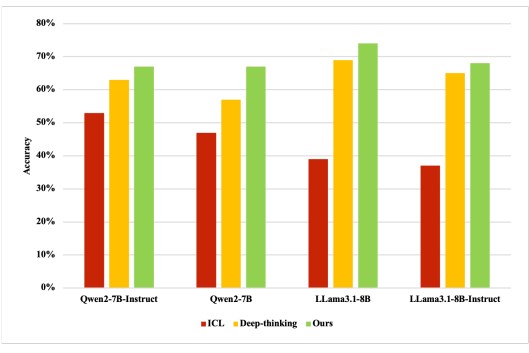

Figure 4: Performance comparison on AE dataset across different ICL enhancement baselines.

## 4.3 MODEL ANALYSIS

**Hyperparameter analysis.** We analyzed the impact of key hyperparameters on model performance, with a particular focus on the total number of samples in the codebook. This refers to the total number of task vectors stored in the codebook during inference. In our approach, the retrieval quantity is fixed at half of the codebook's total storage capacity. Empirical results indicate that as the total number of samples increases, model performance improves steadily across multiple evaluation metrics. This behavior can be attributed to a larger pool of task vectors providing more diverse interpretations, thereby enhancing the model's ability to make accurate predictions. However, the relationship

between retrieval quantity and model performance is not strictly linear. Excessive retrieval may lead to computational inefficiencies and potential overfitting to the codebook, highlighting the importance of finding an optimal retrieval size that balances performance gains with computational costs. The results are presented in Figure 5, demonstrating the performance of our method, conventional ICL, and deep threading on AE datasets.

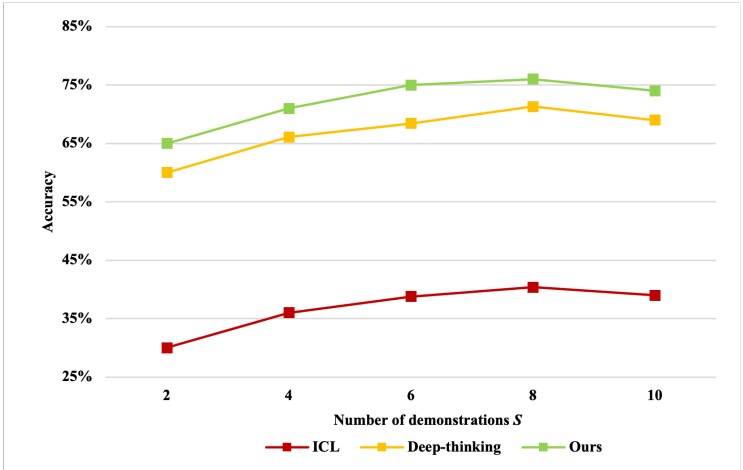

Figure 5: Hyperparameter impact of codebook size $S$ on Llama3.1-8B performance

**Time Complexity.** One potential concern regarding our method is the increased time consumption, primarily due to multiple interpretations of presentations and storing a large number of presentations in the codebook. This could potentially lead to prolonged computation times for LLMs. However, our empirical tests show that the method does not suffer from high time complexity. This is likely because the cost of a single ICL inference is relatively low, and the additional computational overhead introduced by our approach is minimal. Table 3 compares the time consumption of conventional ICL and our method under different quantization settings, while Table 4 provides detailed time consumption in seconds.

Table 3: Time consumption comparison of conventional ICL and ours under different settings.

| Model & Method | Quantization | Time (min) |
|---|---|---|
| ICL (Qwen2-7B) | $N$ | ˜40 min |
| ICL (Llama3.1-8B) | $Y$ | ˜20 min |
| Ours (Llama3.1-8B) | $Y$ | ˜50 min |

Table 4: Detailed time consumption (in seconds) for conventional ICL and ours.

| Model | SST2 | SST5 | MR | AGNews | Average |
|---|---|---|---|---|---|
| OPT-125M | 265.28 | 483.59 | 333.20 | 804.65 | 470 |
| OPT-350M | 627.36 | 1137.13 | 767.53 | 1850.79 | 1095 |
| Qwen2-1.5B | 167.27 | 303.43 | 215.70 | 1087.47 | 443 |
| Qwen2-7B | 452.80 | 824.03 | 762.42 | 4091.94 | 1533 |
| Llama3.1-8B | 929.90 | 1658.05 | 1161.85 | 2835.42 | 1646 |

## 5 CONCLUSION

In this paper, we introduced a novel method for enhancing ICL in LLMs by leveraging a retrieval-based codebook mechanism. Our approach addresses two key challenges in ICL: optimizing the use of key-value pairs within the transformer architecture for enhanced contextual understanding and mitigating input length constraints and noise through efficient task vector storage and retrieval. By dynamically storing and updating historical task vectors in the codebook, our method allows for the retrieval of only the most pertinent information during inference, significantly improving model accuracy and efficiency. Empirical evaluations on widely used datasets, as well as an internal dataset, demonstrated that our approach consistently outperforms conventional ICL, particularly in LLMs with smaller parameter sizes. Furthermore, our analysis of hyperparameters highlights the importance of balancing codebook size to maximize performance gains while minimizing computational overhead. The proposed method also maintains manageable time complexity, further validating its practical applicability. Our work sets a new benchmark for ICL in LLMs and opens avenues for further exploration of retrieval-based mechanisms and dynamic memory structures to enhance ICL performance. Future research could explore optimizing codebook management, including more advanced strategies for knowledge editing and retrieval, as well as extending the methodology to other downstream tasks and model architectures.

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
