# OpenReview forum: "Boosting In-Context Learning in LLMs with Retrieval-based Codebook"
_ICLR.cc/2025/Conference — Submitted to ICLR 2025_

### Official Review · Reviewer_pQ2m · 2024-11-02

**Soundness:** 2
**Presentation:** 2
**Contribution:** 3
**Rating:** 5
**Confidence:** 3

**Summary:**

Existing ICL methods are highly dependent on example selection, example ordering, and instruction formatting, leading to performance instability. This paper proposes a transformer-based approach for encoding demonstrations to enhance the ICL context. A codebook-based method is introduced to store key-value pairs from demonstrations processed multiple times by the LLM to address length limitations and reduce noise. This codebook retains only information relevant to ICL, ensuring that the most effective memory is retrieved upon input of a test query. Extensive experiments demonstrate that the proposed ICL method outperforms the standard ICL paradigm.

**Strengths:**

S1: This paper proposed a new demonstration retrieval method for ICL from the perspective of representations. Notably, it incorporates both the key-value pairs of transformers across each layer and the raw demonstrations, facilitating a more comprehensive context understanding.
S2: A dynamic codebook method is proposed to update the demonstration information, obtain relevant content, and filter irrelevant knowledge. This method also addressed the limitation imposed by input length constraints.
S3: Experiments on five datasets demonstrate the improvement of the proposed ICL paradigm compared to the existing standard ICL methodology.

**Weaknesses:**

W1: The idea of leveraging key-value pairs in transformers is valuable and well-motivated; however, the experimental evidence is insufficient to comprehensively demonstrate the proposed method's advantages. Specifically, the results are not fully convincing in substantiating the method’s superiority, as suggested in the introduction.

- Lines 69-70 mention “…making prompt engineering unstable...”, yet no experiments are presented to show that the proposed method enhances ICL stability.
- Table 2 compares the proposed method with only one baseline, omitting other ICL methods such as example selection and example ordering approaches. Including these comparisons would provide a more comprehensive evaluation.
- Adding several ablated versions of the proposed method in Table 2 would enhance the analysis, particularly by examining the effects of considering only partial layers of transformers and observing corresponding performance changes.
- A brief description of the deep-thinking method depicted in Figures 4 and 5 would enhance clarity in the experimental settings.
- The results in Tables 3 and 4 are somewhat unclear. It is not explained why 'Ours (Qwen2-7B)' is absent in Table 3, nor how to distinguish the proposed method's results from those of conventional ICL methods in Table 4."
W2: Several details of the proposed algorithm could be refined to improve clarity. For example, in lines 3 and 14 of Algorithm 1, $for n \in N$ should be $for n \in [1, N]$, as N represents a count, not an interval. In line 15 of the Algorithm, the meaning of $CB_{c\cdot}$ is unclear. Adding comments would improve readability.
W3: In Eq.(13), some notations require clarification.
W4: Symbols in figure 3 should be briefly explained in the caption， such as “Transformer Pre-part” and “Transformer Post-Part”.
W5: The settings for the proposed method and baselines should be clarified, including details on demonstration selection for conventional ICL and experimental seed choices. Ensuring fairness in comparative experiments is essential.
W6: The citations of previous work in line 207 should be added and Equations in line 278 should be referred by “\ref{}”. The reference format needs correction, e.g., 'Yang et al. (2024b)' in line 113 should appear as '(Yang et al., 2024b)'.

**Questions:**

Is the use of a codebook more like an engineering implementation rather than a theoretical innovation?

---

### Official Review · Reviewer_8wJm · 2024-11-02

**Soundness:** 2
**Presentation:** 1
**Contribution:** 2
**Rating:** 3
**Confidence:** 5

**Summary:**

In this paper, the authors propose a novel in-context learning (ICL) method for large language models (LLMs). The core assumption of the method is that the presentation order of demonstration examples—encapsulated as keys and values within the transformer blocks—is less critical. The authors convert these demonstrations into task vectors, which are further enhanced by stacking several LLMs. To address the limitation of input length constraints, they construct a codebook from which only the most similar task vectors are retrieved during inference. The retrieved results are merged and then used to obtain the final ICL outputs. The authors conduct experiments on multiple public datasets and one in-house dataset, demonstrating that their method outperforms baseline approaches. They also show that the relationship between codebook size and optimal performance is not strictly linear.

**Strengths:**

- S1: The paper provides a thorough discussion of the limitations of current ICL approaches, particularly with respect to prompt sensitivity and input length constraints, and proposes an alternative method to address these issues.
- S2: The introduction of a codebook to tackle input length constraints is well-motivated and adds an innovative element to ICL.
- S3: The method’s effectiveness is demonstrated on multiple public datasets, as well as a proprietary dataset, showcasing the method’s robustness.

**Weaknesses:**

- W1: Writing and technical clarity need improvement. Certain details, such as the construction of initial samples for the codebook (e.g., eq 8), are inadequately explained.
- W2: Comparisons with existing ICL methods (such as [1-3]) are missing, which would have provided a more comprehensive view of the method's relative performance.
- W3: The inference time for the proposed method, as shown in Table 3, significantly increases compared to baselines.
- W4: The authors’ assumption that “the presentation order may be less critical when demonstrations are fully encapsulated within the keys and values across LLM layers” is not empirically validated. Testing the impact of shuffling the retrieved codebook entries could help verify this assumption.
- W5: The impact of the retrieval algorithm for the codebook on overall performance is not studied, leaving questions about the sensitivity of the method to retrieval quality.

[1] Batch-ICL: Effective, Efficient, and Order-Agnostic In-Context Learning, ACL 2024

[2] Towards Informative Few-Shot Prompt with Maximum Information Gain for In-Context Learning, Findings of EMLP 2023

[3] Not All Demonstration Examples are Equally Beneficial: Reweighting Demonstration Examples for In-Context Learning, Findings of EMLP 2023

**Questions:**

- Q1: How does the order of LLMs in eq (7) impact performance?
- Q2: How are initial samples for the codebook initialized in eq (8)?
- Q3: What fusion method was applied in the experiments for combining retrieved entries?
- Q4: Line 261 mentions stacking several LLMs, yet this process is not clearly described in Algorithm 1. Could this be clarified?


Minor errors:
- Line 291: the symbols here are for demonstration examples instead of test input.

---

### Official Review · Reviewer_WqRm · 2024-11-02

**Soundness:** 4
**Presentation:** 3
**Contribution:** 3
**Rating:** 8
**Confidence:** 4

**Summary:**

This paper presents a new approach to enhancing in-context learning (ICL) in large language models (LLMs) through a retrieval-based codebook. The authors propose converting demonstration data into "task vectors" in the form of key-value pairs and storing these in a dynamic codebook. This codebook enables efficient retrieval of pertinent historical data and optimizes the contextual understanding of the LLM. The main contributions are twofold: improving the efficiency and accuracy of ICL by dynamically updating task vectors within the codebook and mitigating input length constraints by filtering irrelevant information. Experimental results demonstrate that the proposed method significantly outperforms conventional ICL, particularly in low-resource settings, and maintains reasonable time complexity, setting a new benchmark for ICL optimizations in LLMs.

**Strengths:**

1. The retrieval-based codebook is a novel application in ICL, introducing an effective way to manage and utilize task vectors for enhanced contextual learning.
2. The experiments span several datasets and LLM architectures, providing robust evidence of the approach’s effectiveness.
3. The codebook mechanism not only enhances accuracy but also addresses computational efficiency, making it feasible for application in real-world, long-context tasks.

**Weaknesses:**

1. The paper’s theoretical sections, particularly around the retrieval and fusion of task vectors, are dense and could be simplified for accessibility.
2. While the approach improves ICL efficiency, the iterative task vector enhancement and codebook retrieval may introduce additional computational costs.

**Questions:**

How does the additional computational overhead from the iterative task vector refinement impact model inference time, particularly for large-scale tasks?

---

### Official Review · Reviewer_3SUk · 2024-11-03

**Soundness:** 1
**Presentation:** 1
**Contribution:** 2
**Rating:** 3
**Confidence:** 5

**Summary:**

This paper introduces a novel approach to demonstration selection by proposing a retrieval-based strategy for choosing the most appropriate demonstration. Specifically, it leverages a deep-thinking method to create condensed Key-Value (KV) representations, facilitating more effective demonstration retrieval.

The contributions of this paper are as follows:
A retrieval-based approach for demonstration selection employing deep-thinking is proposed.

**Strengths:**

1. The paper reinterprets the deep-thinking method from the perspective of task vectors and learning algorithms, providing valuable insights.
2. It extends the deep-thinking approach to test inputs, enhancing the representation’s precision.
3. The proposed retrieval-based method effectively identifies suitable demonstrations.

**Weaknesses:**

Method:
The novelty of the approach appears limited, as it mainly extends existing deepthinking methods by incorporating a retrieval-based mechanism. Additionally, the reasoning behind certain design choices—such as the fusion technique, retrieval strategy, and the use of Key values for retrieval—was not provided, which limits the methodological clarity.

Evaluations:
The authors evaluated their methodology on classification tasks but not to demonstrate its effectiveness across a diverse set of tasks, including NLI. Moreover, the paper lacks a direct comparison with the original deep-thinking approach and other ICL methods. The authors claim to address issues related to handling long inputs and prompt design, yet there are no experiments provided to substantiate these claims.

Writing:
The paper does not provide an adequate explanation of the original deep-thinking method, making it challenging to fully understand the proposed methodology from this paper alone. Furthermore, there are inconsistencies among Algorithm 1, Figure 2, and the text description. For example, Figure 2 does not depict the multiple boosting of test inputs, and there is an error in the initialization step of the pseudo code in Algorithm 1.

**Questions:**

1. Is there a performance difference between the original deep-thinking approach and the retrieval-based approach presented here?
2. Could you provide an ablation study on the fusion and retrieval strategies?
3. What is the specific cause of the performance improvement? Additionally, could you provide quantitative and qualitative examples showing the selected demonstrations and whether retrieval reduces noise?
4. Compared to other demonstration selection methods, what are the advantages of this retrieval-based approach?

---

### Meta-Review · Area_Chair_Ftyh · 2024-12-15

**Metareview:**

In this paper, the authors proposed a new ICL method by selecting demonstrations based on the idea of deep thinking.

There are several major concerns raised by the reviewers: 1, the technical novelty of the proposed method is limited, compared with existing techniques; 2, the experimental results are not convincing --- there is insufficient comparison with SOTA and related ICL methods, and some important claims are not supported by experiments.

The authors did not respond to the reviewers' concerns/questions during the rebuttal.

As the concerns raised by the reviewers are not addressed, this paper is not ready for publication.

**Additional Comments On Reviewer Discussion:**

The authors did not respond to the reviewers' concerns/questions.

---

### Decision · Program_Chairs · 2025-01-22

Reject